# Mass-Spectrometry-Based Lipidomics Discriminates Specific Changes in Lipid Classes in Healthy and Dyslipidemic Adults

**DOI:** 10.3390/metabo13020222

**Published:** 2023-02-03

**Authors:** Salvador Sánchez-Vinces, Pedro Henrique Dias Garcia, Alex Ap. Rosini Silva, Anna Maria Alves de Piloto Fernandes, Joyce Aparecida Barreto, Gustavo Henrique Bueno Duarte, Marcia Aparecida Antonio, Alexander Birbrair, Andreia M. Porcari, Patricia de Oliveira Carvalho

**Affiliations:** 1Health Sciences Postgraduate Program, São Francisco University–USF, Bragança Paulista 12900-000, SP, Brazil; 2Integrated Unit of Pharmacology and Gastroenterology (UNIFAG), São Francisco University–USF, Bragança Paulista 12900-000, SP, Brazil; 3Department of Pathology, Federal University of Minas Gerais, Belo Horizonte 31270-901, MG, Brazil; 4Department of Dermatology, School of Medicine and Public Health, University of Wisconsin-Madison, Madison, WI 53715-1149, USA; 5Department of Radiology, Columbia University Medical Center, New York, NY 10032, USA

**Keywords:** dyslipidemia, lipidomics, triacylglycerols, cholesterol lipoproteins, mass spectrometry

## Abstract

Triacylglycerols (TAGs) and cholesterol lipoprotein levels are widely used to predict cardiovascular risk and metabolic disorders. The aim of this study is to determine how the comprehensive lipidome (individual molecular lipid species) determined by mass spectrometry is correlated to the serum whole-lipidic profile of adults with different lipidemic conditions. The study included samples from 128 adults of both sexes, and they were separated into four groups according to their lipid profile: Group I—normolipidemic (TAG < 150 mg/dL, LDL-C < 160 mg/dL and HDL-c > 40 mg/dL); Group II—isolated hypertriglyceridemia (TAG ≥ 150 mg/dL); Group III—isolated hypercholesterolemia (LDL-C ≥ 160 mg/dL) and Group IV—mixed dyslipidemia. An untargeted mass spectrometry (MS)-based approach was applied to determine the lipidomic signature of 32 healthy and 96 dyslipidemic adults. Limma linear regression was used to predict the correlation of serum TAGs and cholesterol lipoprotein levels with the abundance of the identified MS-annotated lipids found in the subgroups of subjects. Serum TAG levels of dyslipidemic adults have a positive correlation with some of the MS-annotated specific TAGs and ceramides (Cer) and a negative correlation with sphingomyelins (SMs). High-density lipoprotein-cholesterol (HDL-C) levels are positively correlated with some groups of glycerophosphocholine, while low-density lipoprotein-cholesterol (LDL-C) has a positive correlation with SMs.

## 1. Introduction

Dyslipidemia is a highly prevalent condition in the adult population [1] and is characterized by abnormally elevated blood concentrations of clinically relevant lipoproteins, such as low-density lipoprotein-cholesterol (LDL-C), high-density lipoprotein-cholesterol (HDL-C), and triacylglycerols (TAGs) [2]. These lipid alterations are frequently associated with increased cardiovascular risk [3] and with the etiopathogenesis of several cardiometabolic disorders, such as diabetes [4], liver steatosis [5], and pancreatitis [6].

Nevertheless, these lipids have fundamental roles in cell functioning and hemostasis fine-tuning. For example, cholesterol is not only a fundamental cell membrane component but also the main precursor for the synthesis of bile acids, steroid hormones and signaling mediators [7,8]. Lipoproteins are responsible for the transportation of cholesterol and TAG and are present as functionally different specimens, varying in size and density, such as chylomicrons (CMs), very low-density lipoprotein (VLDL), low-density lipoprotein (LDL), intermediate-density lipoprotein (IDL) and high-density lipoprotein (HDL) [7]. The importance of these individual lipids and their association has been actively studied to clarify how their metabolism is related to pathological conditions [5,9,10]. Notably, LDL-C has well-established causal relationships with atherosclerotic cardiovascular disease (ASCVD), where higher levels of LDL-C predict increased ASCVD risk, probably through pro-oxidative and pro-inflammatory mechanisms [11]. TAG, in turn, also presents an indirect role in pathogenesis associated with atherosclerosis, whereas other components related to it, such as the cholesterol esters of TAG-rich lipoproteins, are also involved in ASCVD risk and are correlated with overall TAG abundance [12,13,14]. Despite this, it remains unknown which associations or even whether causative relationships exist between the altered levels of these lipids and their ratio with the emergence of ASCVD and related metabolic disorders [15].

Recent studies have addressed these complex lipid relationships using highly sensitive unbiased methods, such as mass spectrometry (MS) in an untargeted lipidomic approach, building predictive models [16]. In this context, the present study aims to describe possible associations between the clinically relevant serum lipid levels and the lipidomic profiles of healthy and dyslipidemic adults. Additionally, we sought to adjust the effect of well-known inter-subject covariables, such as sex and age, on these correlation outcomes. For this purpose, we used limma [17], a linear regression model developed for transcriptomics data analysis that relates individual variables (features/genes) and phenotypic variables and considers the fixed and random effects of covariates.

Using ultra-high-performance liquid chromatography coupled to electrospray ionization quadrupole time-of-flight mass spectrometry operating in high energy collision spectral acquisition mode (MS^E^) mode (UPLC-QTOF-MS^E^) and avoiding a subjective cut-off of clinical classification, we found groups of MS-annotated lipids that linearly change (after covariate adjustment), in positive or negative ratios, for a broad window of serum TAG, HDL-C, and LDL-C levels.

## 2. Experimental Design

### 2.1. Study Design, Participants and Sampling

An observational, cross-sectional study, with convenience sampling, was carried out to obtain serum samples from adults admitted for laboratory exams at the Integrated Unit of Pharmacology and Gastroenterology (UNIFAG) in the city of Bragança Paulista (São Paulo, Brazil). The study included samples from 128 adults of both sexes, between 18 and 87 years old. The adults were classified into 4 clinical groups (Table 1) according to their lipid profile: normolipidemic (TAG < 150 mg/dL, LDL-C < 160 mg/dL and HDL-c > 40 mg/dL); isolated hypertriglyceridemia (TAG ≥ 150 mg/dL); isolated hypercholesterolemia (LDL-C ≥ 160 mg/dL) and mixed dyslipidemias—combinations of increased LDL-C and TAG (LDL-C ≥ 160 mg/dL and TAG ≥ 150 mg/dL). Adult volunteers were selected and screened based on hemato-biochemical analysis and clinical examination. The exclusion criteria were current smoking, obesity (body mass index (BMI) > 30 kg/m^2^), regular use of any drugs or supplements, congenital heart disease history, diabetes, hypothyroidism, or any other chronic disease.

We used data from adults from all the groups to identify the relationship between known serum lipid levels and lipidomic profiles (MS-annotated lipids) across these 4 clinical classifications. Additional clinical data of subjects are presented in Appendix A.

Peripheral venous blood samples were collected into vacuum blood collection tubes. The serum samples were obtained by centrifugation of the blood samples (1500× *g* for 15 min at 4 °C) immediately after being drawn and then stored at −70 °C until the analysis. Determination of total serum cholesterol (TC), HDL-C, and TAG in fasting blood samples were performed by enzymatic colorimetric methods (Roche Diagnostics, Mannheim, Germany), and LDL-C was calculated using the Friedewald formula [18]. Dyslipidemias were classified according to criteria indicated by the Brazilian Dyslipidemia Directive [19].

### 2.2. Mass Spectrometry Analysis

As previously described by our group [20,21], serum samples (800 μL) were extracted with 2.5 mL of a mixture of chloroform-methanol 2:1 (% *v*/*v*) and 0.5 mL of an aqueous solution of NaCl (0.1 M). The samples were then centrifuged (10,000× *g* for 15 min at 4 °C), and the supernatant was collected and dried under N_2_ gas flow. An equal aliquot of each sample was collected to compose a pooled sample used as quality control (QC). The untargeted lipidomic analysis was performed using an ACQUITY UPLC coupled to a XEVO-G2XS QTOF mass spectrometer (Waters, Manchester, UK). Mobile phase A was composed of a solution of 10 mM ammonium formate with 0.1% formic acid in acetonitrile (ACN) and water (60:40, % *v*/*v*), while mobile phase B was composed of a solution of 10 mM ammonium formate with 0.1% formic acid in isopropanol and ACN (90:10, *v*/*v*). The injection volume was 2 μL through an Acquity UPLC CSHC18 column (2.1 × 100 mm, 1.7 μm, Waters). The flow rate was 0.4 mL min^−1^. The column was initially eluted with 40% B, increasing to 43% B during 0.5 min and subsequently to 63% within 0.1 min. Over the next 3.4 min, the gradient was further ramped to 68% B and then to 80% of B in 0.1 min. Over the next 2.4 min, the gradient was further ramped to 85% B and then to 99% of B in 1.5 min. In the final part of the gradient, the amount of B remained at 99% for 1 min and then returned to 40% in 0.1 min for column re-equilibration during the next 1.9 min. The total run time was 10 min. We separately recorded positive (+) and negative (−) ion modes in the range of 50–1500 *m*/*z*, with an acquisition time of 0.1 s per scan. Other parameters were as follows: source temperature = 140 °C, desolvation temperature = 550 °C (+) and 400 °C (−), desolvation gas flow = 900 L h^−1^, capillary tension = 3.5 kV (+)/2.5 kV (−), cone voltage = 40 V. Sample injection order was random, and QC samples were included after every ten injections. Leucine enkephalin (molecular weight = 555.62; 200 pg μL^−1^ in 1:1 ACN:H_2_O) was used as a lock mass for accurate mass measurement.

### 2.3. Data Analysis and Feature Selection

Progenesis QI 2.0 software (Nonlinear Dynamics, Newcastle, UK) was used for peak alignment, deconvolution, the selection of possible adducts, and compound annotation based on data-independent acquisition—MSE.Ion abundance data were corrected by the QC pool using the QC-RFSC (QC-based random forest signal correction) method implemented in the Systematic Error Removal using the Random Forest (SERRF) package [22]. Preprocessed data are presented in Appendix A. Appendix A show the PCA plots and QC pool samples for negative and positive ion modes, respectively.

Interquartile range (IQR) and QC relative standard deviation (RSD%) were used to filter out non-informative and confounding features. Peaks present in more than 80% of the samples of each group were kept for further analysis. The corrected data were log-transformed and normalized using the Pareto scale.

For feature selection, limma regression implemented in MetaboAnalystR 3.0 [23] was used to individually model TAG, TC, HDL-C, and LDL-C as phenotype variables. Sex and age data were considered covariates and used to correct the linear model. The features with FDR-adjusted *p*-values < 0.05 were kept for downstream analysis, and the obtained coefficients were used to quantify the relationship between features and serum lipids.

For lipid annotation of relevant features, we accepted a precursor mass error of <5 ppm and a fragmentation score of <10 ppm, and the isotopic similarity was evaluated from a list of suggested putative identifications from Progenesis QI software using the LIPID MAPS [24] database and the Human Metabolome Database (HMDB) [25].

## 3. Results

The processing of Raw LC-MS data resulted in 2618 *features* (+)/2618 *features* (−). The selection by regression model resulted in 256 features 140 (+)/116(−), and of those, 36 were identified (11 features (+)/25 features (−)). Limma linear regression models were implemented for each of the three serum lipid markers (TAG, HDL-C and LDL-C) and for total cholesterol (TC), as quantified in the serum in the dyslipidemic adults. For each model, a list of statistically significant features (*retention time and m/z- rt_mz*) was obtained, and, where applicable, as described in the Methods section, lipids were annotated. The results are presented in Appendix A. Figure 1 summarizes the results from all four models. This heatmap shows the coefficients of serum lipids for each MS-annotated lipid. These coefficients represent how much the MS-annotated lipid abundance, after covariate adjustment, changes per unit of the phenotype serum lipid for which it was modeled. The sign of the coefficients indicates whether the association is direct (+) or inverse (−). Complete statistical and chemical data are presented in Appendix A.

### 3.1. MS-Annotated Long-Chain TAG and Cer, Are Positively Associated with TAG Serum Levels

Lipidomics identified some long-chain TAGs, mostly unsaturated (TAG 46:2, TAG 46:1, TAG 48:2, TAG 50:2, TAG 52:2, TAG 54:2, TAG 52:1, TAG 50:0), that are positively correlated to TAG levels in the blood. Long-chain unsaturated dihydroceramides (Cer 38:1;O2, Cer 40:2;O2, Cer 41:2;O2, Cer 42:2;O2, Cer 40:1;O2) show a positive association with serum TAG, while some saturated SMs show a negative association (SM 34:1;O2, SM 38:2;O2). Finally, there is no coordinated trend in glycerophospholipids correlated to serum TAG. Table 2 summarizes characteristics of MS-annotated lipids with statistically significant associations with serum TAG levels (Figure 2).

### 3.2. MS-Annotated PC Associated with Serum HDL-C Levels

There is a marked positive association between serum HDL-C and some molecules of unsaturated PCs (PC 38:7, PC 34:3, PC 40:5, PC 34:1). Table 3 summarizes the characteristics of MS-annotated lipids with statistically significant associations with serum HDL-C levels (Figure 3).

### 3.3. MS-Annotated SM Associated with Serum LDL-C Levels

A set of SMs (SM 35:1;O2, SM 36:1;O2, SM 42:2;O2, SM 40:1;O2, SM 41:1;O2, SM 42:1;O2) was positively associated with serum LDL-C levels. This set shows a different pattern from the others correlated to TAG levels. Two of these SMs are positively correlated to total cholesterol, even when the levels of all lipoproteins differ between hypercholesterolemic and normolipidemic adults. Table 4 summarizes the characteristics of MS-annotated lipids with statistically significant associations with serum LDL-C levels (Figure 4).

## 4. Discussion

Clinically, high levels of TAG result in the accumulation of fat, with increased inflammatory, cardiovascular, and diabetic risk [28,29]. Here, we have identified a set of MS-annotated, long-chain TAGs whose relative abundance maintains a correlation with serum TAG, even for different dyslipidemic profiles, and is not quantitatively correlated to serum HDL or LDL levels, in which there is a predominance of other classes of lipids.

Studies in animal models have indicated that elevated SMs levels are associated with insulin resistance by inducing mitochondrial dysfunction, reactive oxygen species production, and inflammation [30], while increased Cer levels dysregulate glucose homeostasis and accelerate Type 2 diabetes progression [31,32]. Previously, positive associations between unsaturated SMs (C34:1, C36:1, C42:3), hydroxyl-SMs (C34:1, C38:3), hexosylceramide (d18:1/20:1) and acylcarnitines have also been described in plasma; they are positively associated with incident Type 2 diabetes and may reflect dysregulated fatty acid oxidation and mitochondrial stress [32]. The distinct fatty acid composition in Cer may also reflect substrate abundance from diet during Cer synthesis. For example, monounsaturated Cer levels are significantly associated with an elevated risk of Type 2 diabetes in Chinese populations [32] rather than saturated Cer levels, which are reported in Western diet-based populations [33].

In this study, we found a group of non-hydroxy fatty acid dihydrosphingosine (Cer 38:1;O2, Cer 40:2;O2, Cer 41:2;O2, Cer 42:2;O2 and Cer 40:1;O2) positively correlated to serum TAG, probably to compensate for the required increase of dihydroceramide and the DhCer/Cer ratio necessary for TAG-rich VLDL synthesis, as observed by a previous study [34]. Additionally, the annotated non-hydroxy fatty acid sphingosine (Cer 42:3;O2) was positively associated with LDL-C. In agreement with our findings, LDL-C concentrations were strongly and positively correlated with concentrations of the majority of SM, Cer and Hex-Cer species, as well as dihydroceramide [35]. Sphingolipidomics could potentially be used as a tool for the early diagnosis of atherosclerosis in systemic lupus erythematosus [35] as well as other metabolic disorders [31,32] and may have an added benefit to the currently available tools in their diagnosis, prognosis and treatment.

In addition, we identified that PCs are positively correlated to HDL-C levels. A possible explanation for this finding is that PCs can be temporarily sequestered by the apolipoprotein A-I during HDL generation, as previously reported [36]. PC and PE are two major phospholipids that are asymmetrically distributed in the plasma membrane of animal cells, and small alterations in phospholipids levels appear to have large implications for parameters (lipid profiles, obesity, insulin resistance and liver disease progression) related to metabolic disorders. The content of polyunsaturated fatty acids in the phospholipid of skeletal muscle may be important for efficient insulin action [37]. However, the mechanisms by which these phospholipids influence insulin action remain unclear. It is likely that the unbalanced synthesis of muscle PC and PE influences muscle insulin sensitivity by disrupting cellular calcium homeostasis [38]. The appropriate hepatic PC/PE ratio has clinical relevance for liver health since a significant proportion of patients with non-alcoholic fatty liver disease and non-alcoholic steatohepatitis have a lower hepatic PC/PE ratio compared to control livers [39]. Modulation of PC/PE levels suggests that this ratio is a key regulator of cell membrane integrity and plays a role in the progression of steatosis into steatohepatitis [40]. Furthermore, a positive association between plasmalogen-PCs and circulating HDL has been previously reported, which contributes to the atheroprotective properties of HDL. Localized increases of plasmalogen-PCs contribute positively towards the cardioprotective properties of HDLs, including the elevated capacity of cholesterol efflux from endothelial cells, anti-apoptotic effects, and an enhanced capacity to halt the propagation of phospholipid hydroperoxide radicals in the surrounding vasculature [41].

## 5. Conclusions

Our findings evidence that changes in individual molecular lipid species are discriminatory for a number of different dyslipidemic conditions. Predicting the correlation of common serum lipids with the lipidome can contribute to identifying potential molecular signatures, which can, ultimately, be used to screen subjects with increased risk for cardiometabolic disorders. The determination of the plasma baseline levels of long-chain (dihydro)ceramides can be a useful parameter to be used clinically in the diagnosis of dyslipidemia as well as for cardiometabolic risk stratification, as previous studies have pointed out. Finding specific molecular correlations to dyslipidemias can also guide the further understanding of the metabolic mechanisms underlying alterations associated with these phenotypes. This demonstrates the potential clinical benefit in determining a patient’s risk of disease that a more detailed lipidomic analysis can provide. Among the limitations of the study are its reliance on confirming the lipid MS-annotation by targeted analysis, not correlating the lipidic profile with other relevant clinical variables (i.e., glucose levels, arterial blood pressure, and abdominal circumference) and not validating the possible biological role of the annotated lipids.

## Figures and Tables

**Figure 1 metabolites-13-00222-f001:**
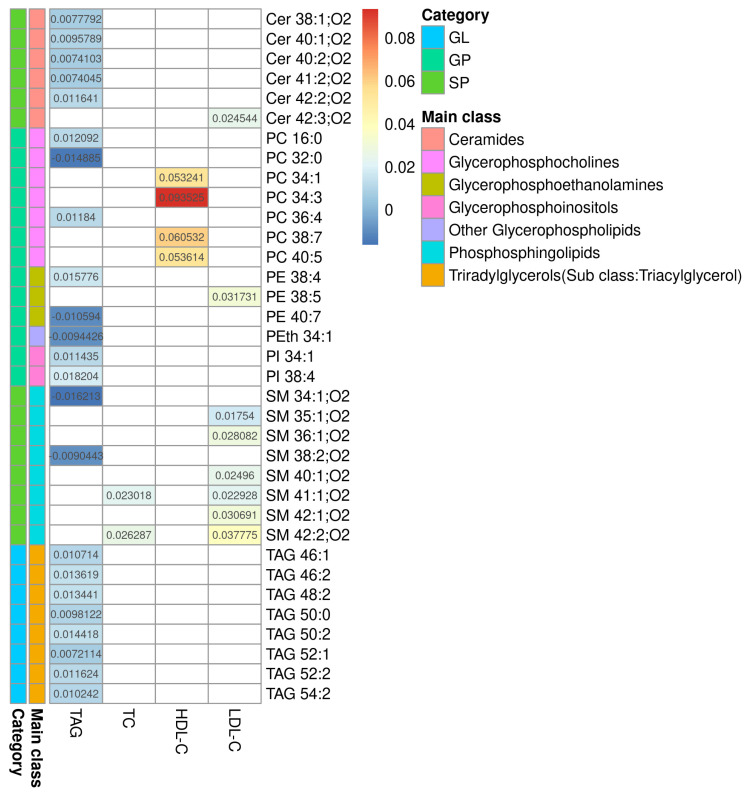
Heatmap of the regression coefficients for the models correlating with TAG, TC, HDL-C, and LDL-C. Heatmap of the coefficients for serum lipids obtained by each model and their MS-annotated lipids. Biochemically determined lipids are shown in columns and MS-annotated lipids are in rows. The colored bars represent the category and main lipid classes, as shown in the legend. GL: glycerolipids; GP: glycerophospholipids; SP: sphingolipids, Cer: ceramide; PC: phosphatidylcholine; PE: phosphatidylethanolamine; PI: phosphatidylinositol; PS: phosphatidylserine; SM: sphingomyelin; TAG: triacylglycerol; TC: total cholesterol; HDL-C: high-density lipoprotein-cholesterol; LDL-C: low-density lipoprotein-cholesterol.

**Figure 2 metabolites-13-00222-f002:**
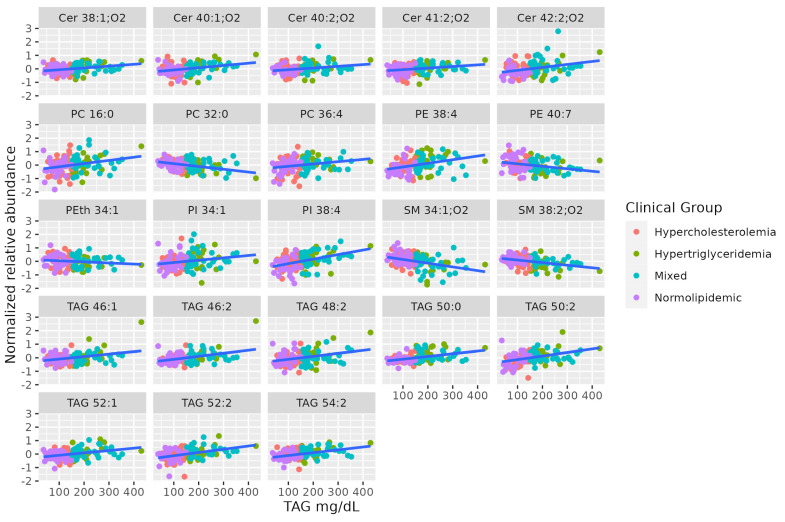
Dot-plot representing the correlation profile between serum TAG levels and the MS-annotated lipid classes. A line obtained by simple linear regression was added to aid visualization.

**Figure 3 metabolites-13-00222-f003:**
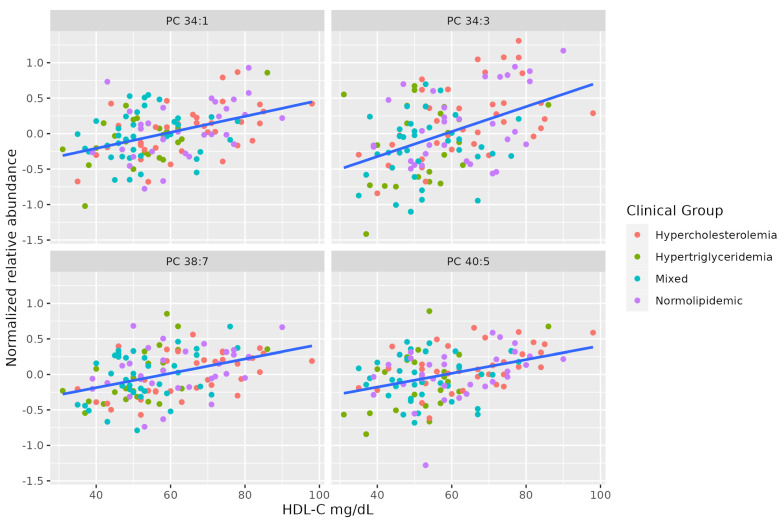
Dot-plot representing the correlation between serum HDL-C levels and the MS-annotated lipids. A line obtained by simple linear regression was added to aid visualization.

**Figure 4 metabolites-13-00222-f004:**
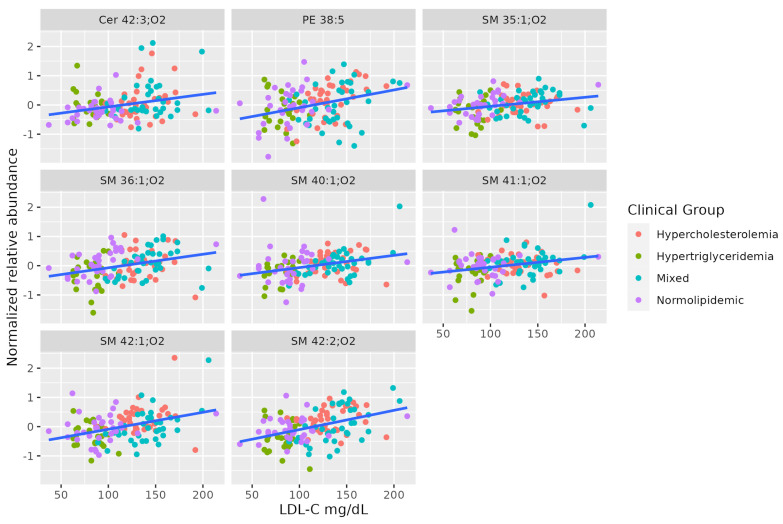
Dot-plot representing the correlation between serum LDL-C abundance and MS-annotated lipids. A line obtained by simple linear regression was added to aid visualization.

**Table 1 metabolites-13-00222-t001:** Clinical classification of subjects according to their biochemical lipid profile. Data are shown as mean (standard deviation).

	n	TAG (mg dL^−1^)	TC (mg dL^−1^)	HDL-C (mg dL^−1^)	LDL-C (mg dL^−1^)
**Normolipidemic**	32	96.68 (28.76)	190.73 (28.6)	63.92 (13.67)	107.32 (26.56)
**Isolated hypertriglyceridemia**	23	211.2 (58.58)	205.15 (30.37)	52.87 (10.83)	111.24 (30.26)
**Isolated hypercholesterolemia**	36	112.33 (25.67)	223.5 (59.74)	57.33 (17.18)	178.33 (20.75)
**Mixed dyslipidemia**	37	213.4 (60.05)	274.8 (25.45)	51.6 (10.83)	183.4 (17.84)

Legend: TAG: triacylglycerol; TC: total cholesterol; HDL-C: high-density lipoprotein-cholesterol; LDL-C: low-density lipoprotein-cholesterol.

**Table 2 metabolites-13-00222-t002:** MS-annotated lipids (rt_*m*/*z*) with statistically significant associations with serum TAG levels.

rt_*m*/*z*	Coefficient TAG	Adj.P. Value	Annotation	Mass Error (ppm)	Matched Fragments	Annotation Confidence Level *
2.82_720.5891 *m*/*z*	−0.01	0.00	PC 32:0	−1.43	537.5244 ^a^	2
0.70_495.3337 n	0.01	0.02	PC 16:0	2.49	184.0744 ^a^	2
2.33_747.5641 *m*/*z*	−0.02	0.00	SM 34:1;O2	−2.35	687.5428 ^a^	2
2.33_766.5372 *m*/*z*	0.01	0.02	PE 38:4	−2.59	303.2329 ^b^	2
2.33_826.5580 *m*/*z*	0.01	0.01	PC 36:4	−3.02	766.5370 ^a^	2
2.55_835.5318 *m*/*z*	0.01	0.03	PI 34:1	−2.93	581.3099/241.0119 ^a^	2
2.56_885.5479 *m*/*z*	0.02	0.00	PI 38:4	−2.19	581.3092/241.0119 ^a^	2
2.57_734.5702 *m*/*z*	−0.01	0.01	PEth 35:1	1.03	383.3521 ^a^	2
2.67_801.6098 *m*/*z*	−0.01	0.01	SM 38:2;O2	−3.93	741.5892 ^a^	2
2.74_766.5376 *m*/*z*	0.02	0.00	PE 38:4	−2.13	303.2329 ^b^	2
2.78_774.5420 *m*/*z*	−0.01	0.02	PE 40:7	−2.99	305.2482 ^b^	2
3.47_638.5728 *m*/*z*	0.01	0.01	Cer 38:1;O2	−0.14	592.5703 ^a^	2
3.50_664.5877 *m*/*z*	0.01	0.03	Cer 40:2;O2	−1.43	618.5831 ^a^	2
3.76_678.6035 *m*/*z*	0.01	0.02	Cer 41:2;O2	−1.16	632.6000 ^a^	2
3.95_692.6179 *m*/*z*	0.01	0.02	Cer 42:2;O2	−2.99	646.6137 ^a^	2
4.01_666.6029 *m*/*z*	0.01	0.01	Cer 40:1;O2	−2.03	620.5986 ^a^	2
5.37_774.6727 n	0.01	0.00	TAG 46:2	−1.33	521.4569/519.4422 ^a^	2
5.58_776.6897 n	0.01	0.00	TAG 46:1	0.34	523.4730/521.4577 ^a^	2
5.59_802.7060 n	0.01	0.00	TAG 48:2	1.15	549.4890/547.4738 ^a^	2
5.79_830.7374 n	0.01	0.00	TAG 50:2	1.23	577.5199/549.4895 ^a^	2
6.02_858.7692 n	0.01	0.00	TAG 52:2	1.77	603.5365/577.5206 ^a^	2
6.25_886.7993 n	0.01	0.00	TAG 54:2	0.44	631.5711/605.5506 ^a^	2
6.26_860.7828 n	0.01	0.04	TAG 52:1	−0.59	605.5506/579.5338 ^a^	2
6.27_834.7655 n	0.01	0.01	TAG 50:0	−2.59	579.5338 ^a^	2

Legend: * Annotation confidence level [26]. (1) Reference standard confirmed structure; (2) exact mass, isotopic pattern, retention time and MS/MS spectrum matched to an in-house spectral database or literature spectra; (3) putative ID assignment based only on elemental formula match with exact mass and isotopic pattern; and (4) unknown compound. ^a^ Annotation based on exact mass measurements of precursors and fragments using a high-resolution mass spectrometer. ^b^ Annotation requires the detection of FA-chain-specific fragments [27]. rt_*m*/*z*: feature label using retention time and mass-to-charge ratios. TAG: coefficient for TAG in the limma model; adj.P.Val: FDR-adjusted *p*-values; ppm: parts per million.

**Table 3 metabolites-13-00222-t003:** MS-annotated lipids with statistically significant associations with HDL-C serum levels.

Rt_*m*/*z*	Coefficient HDL-C	Adj.P.Value	Annotation	Mass Error (ppm)	Match Fragments	Annotation Confidence Level *
2.04_848.5460 *m*/*z*	0.06	0.00	PC 38:7	1.55	788.5238/327.2329 ^b^	2
2.52_786.5633 *m*/*z*	0.09	0.00	PC 34:3	−2.89	726.5423 ^a^	2
2.84_866.6248 *m*/*z*	0.05	0.02	PC 40:5	−3.99	806.6031 ^a^	2
2.86_790.5978 *m*/*z*	0.05	0.02	PC 34:1	1.37	730.5785 ^a^	2

Legend: * Annotation confidence level [26]. (1) Reference standard confirmed structure; (2) exact mass, isotopic pattern, retention time and MS/MS spectrum matched to an in-house spectral database or literature spectra; (3) putative ID assignment based only on elemental formula match with exact mass and isotopic pattern; and (4) unknown compound. ^a^ Annotation based on exact mass measurements of precursors and fragments using a high-resolution mass spectrometer. ^b^ Annotation requires the detection of FA-chain-specific fragments [27]. rt_*m*/*z*: feature label using retention time and mass-to-charge ratios. HDL-C: coefficient for HDL-C in the limma model; adj.P.Val: FDR-adjusted *p*-values; ppm: parts per million.

**Table 4 metabolites-13-00222-t004:** MS-annotated lipids with statistically significant associations with LDL-C serum levels.

rt_*m*/*z*	Coefficient LDL-C	Adj.P.Value	Annotation	Mass Error (ppm)	Match Fragments	Annotation Confidence Level *
2.48_761.5795 *m*/*z*	0.02	0.04	SM 35:1;O2	−2.65	701.5586 ^a^	2
2.65_775.5947 *m*/*z*	0.03	0.01	SM 36:1;O2	−3.23	715.5747 ^a^	2
2.91_750.5440 *m*/*z*	0.03	0.03	PE 38:5	−0.38	464.3146/303.2329 ^b^	2
3.43_857.6733 *m*/*z*	0.04	0.00	SM 42:2;O2	−2.47	797.6521 ^a^	2
3.47_690.6029 *m*/*z*	0.02	0.04	Cer 42:3;O2	−2.08	644.5987 ^a^	2
3.47_831.6576 *m*/*z*	0.02	0.03	SM 40:1;O2	−2.62	771.6366 ^a^	2
3.75_845.6731 *m*/*z*	0.02	0.03	SM 41:1;O2	−2.82	785.6521 ^a^	2
4.03_859.6888 *m*/*z*	0.03	0.02	SM 42:1;O2	−2.72	799.6677 ^a^	2

Legend: * Annotation confidence level [26]. (1) Reference standard confirmed structure; (2) exact mass, isotopic pattern, retention time and MS/MS spectrum matched to an in-house spectral database or literature spectra; (3) putative ID assignment based only on elemental formula match with exact mass and isotopic pattern; and (4) unknown compound. ^a^ Annotation based on exact mass measurements of precursors and fragments using a high-resolution mass spectrometer. ^b^ Annotation requires the detection of FA-chain-specific fragments [27]. rt_*m*/*z*: feature label using retention time and mass-to-charge ratio. LDL-C: coefficient for LDL-C in the limma model; adj.P.Val: FDR-adjusted *p*-values; ppm: parts per million.

## Data Availability

The datasets supporting the conclusions of this article are included within the article and its Appendix A.

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
