# Peer review of "Mass-Spectrometry-Based Lipidomics Discriminates Specific Changes in Lipid Classes in Healthy and Dyslipidemic Adults"

_metabolites, 2023, doi:10.3390/metabo13020222_

Round 1

Reviewer 1 Report

The manuscript by Sánchez-Vinces et al. describes the characterization of serum lipids by mass spectrometry and comparison of this characterization with biochemically determined lipid profiles. The results suggest that some lipid levels may be considered markers of dyslipidemia. The manuscript will probably be interesting for readers of Metabolites.

There are some minor issues to be clarified:

1. Line 92, exclusion criteria: the Authors mention "smoking." They should state if it means current smoking or with a smoking history in a lifetime.

2. Lines 109-111: The reviewer cannot grasp what "the cut-off values for age" means. Please clarify. If it means that levels of lipids for classification to the dyslipidemic group were different depending on the patients' age, it is not "in consonance" with the approach taken in this work, where such classification was independent of age. If the Authors mean something different, they should write more details to avoid ambiguity.

3. Some references lack page numbers or paper numbers.

Author Response

The manuscript by Sánchez-Vinces et al. describes the characterization of serum lipids by mass spectrometry and comparison of this characterization with biochemically determined lipid profiles. The results suggest that some lipid levels may be considered markers of dyslipidemia. The manuscript will probably be interesting for readers of Metabolites.

There are some minor issues to be clarified:

  1. Line 92, exclusion criteria: the Authors mention "smoking." They should state if it means current smoking or with a smoking history in a lifetime.

Answer: Thank you for your time and the kindness in the review. This exclusion criterion has been changed to “current smoking” as proposed (Line 93).

  1. Lines 109-111: The reviewer cannot grasp what “the cut-off values for age” means. Please clarify. If it means that levels of lipids for classification to the dyslipidemic group were different depending on the patients' age, it is not "in consonance" with the approach taken in this work, where such classification was independent of age. If the Authors mean something different, they should write more details to avoid ambiguity.

Answer: Thank you for the observation. This cut-off value stands for the Directive values. We changed: “Dyslipidemias were classified biochemically in consonance with the cut-off value for age as indicated by the current Brazilian Dyslipidemia Directive” to “Dyslipidemias were classified according to criteria indicated by the Brazilian Dyslipidemia Directive”

  1. Some references lack page numbers or paper numbers.

Answer: Thank you for the observation, we have added the information missing from the references.

Reviewer 2 Report

The manuscript reports a potentially important and interesting data. However, the paper  is not well written and has severe flaws and notable weaknesses. Therefore, I cannot recommended publishing this article in the present form.

  Specific comments:   Major points   1.Small size of examined groups (23-37) and confounding variables like: a) very different age of examined subjects/ patients (age between 18 -87 years) and b) both sexes included in the study, which may affect serum lipids concentrations and lipid composition do not allow to make clear conclusion.   2. In conclusion authors wrote that obtained results “(…) demonstrates  the potential clinical benefit in determining a patients risk of the disease (…)” , however they do not say which one of tested parameters has an advantage over serum TAG, total cholesterol, HDL-cholesterol and LDL-cholesterol concentrations, commonly used to diagnose dyslipidemia and consequence of dyslipidemia. Please be more specific.   3.Some statements/sentences are confusing. For instance in Abstract (lines 28-29) authors say that  serum TAG concentration have negative correlation with sphingomyelins, but on page 5 (lines 188-189) that SM is positively associated with TAG serum level. Could you explain this discrepancy.   4.For readers of the paper not familiar with MS and with data presented in J Lipid Res 2020, 61, 1539-1555 (including most of medical doctors) traditional names of fatty acids (like: palmitic acids , stearic acid and so on) should be presented. For instance in Fig. 1 or  Table, 2 column with traditional names of fatty acids should be included (not only “Annotation column”)       Minor points:   1. In sentence “(…) but also the main precursor for synthesis steroid hormones(…)”  Page 2 (line 48)  bile acids should be added. (…) (proposed version: but also the main precursor for synthesis bile acids, steroid hormones (…)”   2. In sentence “(…)Cholesterol circulating lipoprotein (…)” (Page 2, line 49) Cholesterol circulating should be omitted. (proposed version : Lipoproteins are responsible  for …)   3. In sentence “(…) Notably LDL and LDL-C (Page 2 , line 55) LDL and should be omitted (proposed version:  Notably  LDL-C have well established …)  

Author Response

The manuscript reports a potentially important and interesting data. However, the paper is not well written and has severe flaws and notable weaknesses. Therefore, I cannot recommended publishing this article in the present form.

Specific comments:

Major points

1.Small size of examined groups (23-37) and confounding variables like: a) very different age of examined subjects/ patients (age between 18 -87 years) and b) both sexes included in the study, which may affect serum lipids concentrations and lipid composition do not allow to make clear conclusion.  

Answer: We would like to thank you for your time and kindness in reviewing our manuscript. The total sample size is 128 adults of both sexes. The examined groups (23 to 37) are clinical groups. As the adopted regression model utilized for all the samples we consider the complete interval of each lipid. The confounding variables were evaluated in the model (covariates effect in limma model – Lines 79-81). The limma model utilized the Empirical Bayes method to estimate the p value on data with bigger variables over samples. This method is widely used and well documented. This information is described in the text, line 70-74: “Also, we sought to adjust the effect of well-known inter-subject covariables, such as sex and age, on these correlation outcomes. For this purpose, we used limma [17], a linear regression model developed for transcriptomics data analysis that relates individual variables (features/genes) and phenotypic variables and considers fixed and random effects of covariates”

  1. In conclusion authors wrote that obtained results “(…) demonstrates the potential clinical benefit in determining a patients risk of the disease (…)” , however they do not say which one of tested parameters has an advantage over serum TAG, total cholesterol, HDL-cholesterol and LDL-cholesterol concentrations, commonly used to diagnose dyslipidemia and consequence of dyslipidemia. Please be more specific. 

Answer: Thanks for the comment. The corrections suggested by the reviewer have been duly made in the body of the text and highlighted in yellow on page 11, line 333.

3.Some statements/sentences are confusing. For instance in Abstract (lines 28-29) authors say that serum TAG concentration have negative correlation with sphingomyelins, but on page 5 (lines 188-189) that SM is positively associated with TAG serum level. Could you explain this discrepancy.

Answer: Thank you for the observation. The title of section 3.1 has been corrected. Serum TAG concentration is positively correlated with the MS-annotated specific TAGs and the negatively correlated with of SMs (Abstract and Table 2).

4.For readers of the paper not familiar with MS and with data presented in J Lipid Res 2020, 61, 1539-1555 (including most of medical doctors) traditional names of fatty acids (like: palmitic acids , stearic acid and so on) should be presented. For instance in Fig. 1 or Table, 2 column with traditional names of fatty acids should be included (not only “Annotation column”)

Answer: Thank you for the observation. The data shown in Fig. 1 and Table, 2 are the identified classes of lipids and not the separate fatty acids. Although these classes contain different types of fatty acids in their composition, it is not possible to say precisely which they are since the level of identification does not allow it. To facilitate the reader's understanding, all lipid classes have been identified in Figure 1.

Minor points:

  1. In sentence “(…) but also the main precursor for synthesis steroid hormones(…)” Page 2 (line 48) bile acidsshould be added. (…) (proposed version: but also the main precursor for synthesis bile acids, steroid hormones (…)”

Answer: Thank you for the observation; the proposed version has been adopted.

  1. In sentence “(…)Cholesterol circulating lipoprotein (…)” (Page 2, line 49) Cholesterol circulatingshould be omitted. (proposed version : Lipoproteins are responsible for …)

Answer: Thank you for the observation; the proposed version has been adopted.

  1. In sentence “(…) Notably LDL and LDL-C (Page 2, line 55) LDL and should be omitted (proposed version: Notably LDL-C have well established …)

Answer:  Thank you for the observation; the proposed version has been adopted.

Reviewer 3 Report

This manuscript studies the correlation between LCMS-based lipidomics with serum whole-lipidic profiles of adults. Multiple preprocessing were used for MS data, such as lipid annotation and ion abundance correction. Correlation and PCA analysis were carried out. Limma linear regression was also used to correlate MS-annotated lipid abundance to TAG and cholesterol lipoprotein levels. This topic is significant for the clinical application of MS-based lipidomics. Detailed data analysis and discussion are provided. This manuscript can be considered for publication after minor revisions.

Minor points:

1. Page 3 Line 131, what do you mean separately recorded positive and negative ion modes? Did you do alternating positive and negative acquisition in one UPLC elution, or did you do positive or negative in 2 separate elutions?

2. Page 4 Line 142, did you use LC retention time in the lipid annotation?

3. Page 4 Line 149, what’s the purpose for these PCA analysis? I recommend discriminant analysis (such as LDA) for multivariate analysis/visualization.

4. Page 4 Line 155. The feature selection is a little confusing. How many features were detected in raw LC-MS data? How many were annotated as lipids? And how many were selected in the regression models?

5. In Figure 1, what’s the meaning of GL, GP and SP in Category?

6. Since the discussion focuses on the limma regression model, model diagnostics needs to be provided, such as goodness of fit.

Author Response

This manuscript studies the correlation between LCMS-based lipidomics with serum whole-lipidic profiles of adults. Multiple preprocessing were used for MS data, such as lipid annotation and ion abundance correction. Correlation and PCA analysis were carried out. Limma linear regression was also used to correlate MS-annotated lipid abundance to TAG and cholesterol lipoprotein levels. This topic is significant for the clinical application of MS-based lipidomics. Detailed data analysis and discussion are provided. This manuscript can be considered for publication after minor revisions.

Minor points:

  1. Page 3 Line 131, what do you mean separately recorded positive and negative ion modes? Did you do alternating positive and negative acquisition in one UPLC elution, or did you do positive or negative in 2 separate elutions?

Answer: We would like to thank you for your time, help and kindness in reviewing our manuscript. We did the alternating acquisition in two separate elutions as shown in Line 133.

  1. Page 4 Line 142, did you use LC retention time in the lipid annotation?

Answer:  Thank you for your observation. We based the identification on the database of spectra banks. With those banks we don’t have access to the used chromatographic information, making it difficult to use retention time as a variable identification.

Also,at Page 4 Line 162, we changed  “(…)  a fragmentation score >0 from a list of (…)” to “(…) a fragmentation score < 10 ppm and the isotopic similarity was evaluated from a list of (…)”.

  1. Page 4 Line 149, what’s the purpose for these PCA analysis? I recommend discriminant analysis (such as LDA) for multivariate analysis/visualization.

Answer:  Thank you for your observation. The PCA is used as a primary exploration. The variance of the components and the dispersion of the samples in the graphs give an idea of the difference between samples, and the proximity of the QC samples indicates the quality of the analysis, since variations in the lipid abundance or in the concentration of samples are visualized as deviations from them. On the other hand, LDA would be used to evaluate the importance of the variables in relation to the groups (without QC samples), which is not the objective in this case.

  1. Page 4 Line 155. The feature selection is a little confusing. How many features were detected in raw LC-MS data? How many were annotated as lipids? And how many were selected in the regression models?

Answer:  Thank you, we really appreciate your observation. We changed the order of the paragraphs at “2.3. Data analysis and feature selection” for an easier comprehension, as we added: “The processing of Raw LC-MS data resulted in 2618 features (+) / 2618 features (-). The selection by Regression Model resulted in 256 features 140 (+) / 116 (-) and, of those, 36 were identified (11 features (+) / 25 features (-)).”to the “3. Results”.

  1. In Figure 1, what’s the meaning of GL, GP and SP in Category?

Answer: Thank you for the observation, the meanings are: SP – Sphingolipids; GP – Glycerophospholipids; GL- Glycerolipids, and they have been  added at the bottom of the figure.

  1. Since the discussion focuses on the limma regression model, model diagnostics needs to be provided, such as goodness of fit.

Answer:  Thank you for the observation. With limma we are modeling thousands (depending on the number of features) of linear models with covariates, so it is difficult to define a measure of goodness of fit, in addition to the fact that the p value itself is adjusted by the Empirical Bayes method, helping in data analysis where the amount of samples is much less than the number of variables. Furthermore, the limma implementation in MetaboAnalystR was used, so we are limited in extracting intermediate results to elaborate further analyszes (e.g. q-q plots). However, the model suggested by limma is robust and very well documented.

Round 2

Reviewer 2 Report

The manuscript has been significantly improved. I have no further comments.